# The Impact of A3AR Antagonism on the Differential Expression of Chemoresistance-Related Genes in Glioblastoma Stem-like Cells

**DOI:** 10.3390/ph17050579

**Published:** 2024-04-30

**Authors:** Liuba Peñate, Diego Carrillo-Beltrán, Carlos Spichiger, Alexei Cuevas-Zhbankova, Ángelo Torres-Arévalo, Pamela Silva, Hans G. Richter, Ángel Ayuso-Sacido, Rody San Martín, Claudia Quezada-Monrás

**Affiliations:** 1Laboratorio de Biología Tumoral, Instituto de Bioquímica y Microbiología, Facultad de Ciencias, Universidad Austral de Chile, Valdivia 5090000, Chile; jcliuba4834@yahoo.com (L.P.); diego.carrillo@uach.cl (D.C.-B.); alexeiczh@gmail.com (A.C.-Z.);; 2Laboratorio de Virología Molecular, Instituto de Bioquímica y Microbiología, Facultad de Ciencias, Universidad Austral de Chile, Valdivia 5090000, Chile; 3Millennium Institute on Immunology and Immunotherapy, Universidad Austral de Chile, Valdivia 5090000, Chile; 4Laboratorio de Biología Molecular Aplicada, Instituto de Bioquímica y Microbiología, Facultad de Ciencias, Universidad Austral de Chile, Valdivia 5090000, Chile; cspichiger@uach.cl; 5Escuela de Medicina Veterinaria, Facultad de Medicina Veterinaria Y Recursos Naturales, Sede Talca, Universidad Santo Tomás, Talca 347-3620, Chile; atorres32@santotomas.cl; 6Laboratorio de Cronobiología del Desarrollo, Instituto de Anatomía, Histología y Patología, Facultad de Medicina, Universidad Austral de Chile, Valdivia 5090000, Chile; hrichter@uach.cl; 7Faculty of Experimental Sciences, Universidad Francisco de Vitoria, 28223 Madrid, Spain; ayusoa@vithas.es; 8Brain Tumour Laboratory, Fundación Vithas, Grupo Hospitales Vithas, 28043 Madrid, Spain; 9Laboratorio de Patología Molecular, Instituto de Bioquímica y Microbiología, Facultad de Ciencias, Universidad Austral de Chile, Valdivia 5090000, Chile

**Keywords:** glioblastoma, glioblastoma stem-like cells, adenosine, chemoresistance

## Abstract

Glioblastoma (GB) is the most aggressive and common primary malignant tumor of the brain and central nervous system. Without treatment, the average patient survival time is about six months, which can be extended to fifteen months with multimodal therapies. The chemoresistance observed in GB is, in part, attributed to the presence of a subpopulation of glioblastoma-like stem cells (GSCs) that are characterized by heightened tumorigenic capacity and chemoresistance. GSCs are situated in hypoxic tumor niches, where they sustain and promote the stem-like phenotype and have also been correlated with high chemoresistance. GSCs have the particularity of generating high levels of extracellular adenosine (ADO), which causes the activation of the A_3_ adenosine receptor (A3AR) with a consequent increase in the expression and activity of genes related to chemoresistance. Therefore, targeting its components is a promising alternative for treating GB. This analysis determined genes that were up- and downregulated due to A3AR blockades under both normoxic and hypoxic conditions. In addition, possible candidates associated with chemoresistance that were positively regulated by hypoxia and negatively regulated by A3AR blockades in the same condition were analyzed. We detected three potential candidate genes that were regulated by the A3AR antagonist MRS1220 under hypoxic conditions: *LIMD1*, *TRIB2*, and *TGFB1*. Finally, the selected markers were correlated with hypoxia-inducible genes and with the expression of adenosine-producing ectonucleotidases. In conclusion, we detected that hypoxic conditions generate extensive differential gene expression in GSCs, increasing the expression of genes associated with chemoresistance. Furthermore, we observed that MRS1220 could regulate the expression of *LIMD1*, *TRIB2*, and *TGFB1*, which are involved in chemoresistance and correlate with a poor prognosis, hypoxia, and purinergic signaling.

## 1. Introduction

Glioblastoma (GB), also called grade IV astrocytoma, is the most common primary malignant tumor of the central nervous system (comprising 48.6% of malignant tumors of the central nervous system), being the most prevalent among gliomas (57.7%) [1]. GB is more common in older adults and less common in children [2]. The average age of GB diagnosis of 47.9% of patients is ≥65 years, and 46.3% are between 40 and 64 years old [3]. Only 3% of childhood brain tumors are glioblastomas [4]. Unfortunately, the prognosis and life expectancy of patients with GB are dismal [5]. The reported median survival rate is 9.2 months [6], with only 5% of patients surviving more than 5 years [7]. Many attempts have been made to improve or cure this disease, from conventional treatments (surgery, radiotherapy, and temozolomide as a principal chemotherapeutic agent) to modern therapies, including targeted therapy. However, there is still no curative treatment for this pathology, and although conventional treatment provides some improvement, it is far from increasing the survival rate [8,9]. Therapeutic failure in GB can be attributed to several factors, with chemoresistance being the main contributor [10,11]. The presence of a subpopulation of glioblastoma-like stem cells (GSCs) is mainly responsible for generating drug resistance [12] in addition to contributing to tumor recurrence due to their self-renewal capacity [13]. GSCs are located in hypoxic tumor niches that maintain and promote the stem-like phenotype and have also been correlated with high chemoresistance [14].

GSCs are characterized by generating high levels of extracellular adenosine (ADO) through the expression and activity of *NT5E* (CD73) and *ACP3* (PAP) ectonucleotidases [15,16,17]. In GSCs, the extracellular ADO activates the A_3_ adenosine receptor (A3AR) and ultimately increases the expression of genes related to chemoresistance to antitumor drugs. In this regard, an attempt has been made to elucidate the role of MRS1220, a highly selective antagonist for human A3AR [18]. In the present study, RNA sequencing (RNAseq) transcriptomic analysis of GSCs treated with MRS1220 was implemented to identify differentially regulated genes under normoxic and hypoxic conditions. Subsequently, those genes related to chemoresistance were analyzed and identified as potential therapeutic targets that respond to MRS1220 under hypoxic conditions.

Our study identified three chemoresistance-associated genes, *LIMD1*, *TRIB2*, and *TGFB1*, that were top-downregulated by MRS1220 under hypoxic conditions in GSCs. *LIMD1* and *TRIB2* are poorly characterized genes in glioblastoma, and regulation mediated by the A3AR has not been reported. On the contrary, changes in the expression of *TGFB1* have been widely described in glioblastoma. However, it is known that a modification in conjunction with the other genes regulated by MRS1220 could bring benefits in future combined therapy. Through analysis of RNAseq data from GB patients in the GlioVis database, we detected that the expression of these genes is positively correlated with a poor prognosis and with the expression of hypoxia-inducible genes and ectonucleotidase transcripts associated with the purinergic signaling that regulates ADO production and A3AR activation.

## 2. Results

### 2.1. The Effect of MRS1220 on GSC-U87 Gene Expression Cultured under Normoxic versus Hypoxic Conditions

We first identified genes that were differentially regulated under hypoxic (H) versus normoxic (N) conditions from data obtained by RNAseq from U87-GSCs. We showed the primary genes that were regulated by the hypoxic condition in a heat map considering the top up- and downregulated genes (Figure 1A). In this case, 42,455 genes were analyzed, of which 17,459 were identified, with 2956 (17%) upregulated genes and 3680 (21%) downregulated (Figure 1B). As in the previous comparison, the primarily regulated genes comparing normoxia MRS1220 (N*) versus hypoxia MRS1220 (H*) were added to a heat map considering the top upregulated and downregulated genes (Figure 1C). In this comparison, 17,983 genes were analyzed; of these, 3328 (18%) were positively regulated, and 3942 (22%) were negatively regulated (Figure 1D). The data obtained from the volcano plot were considered for both comparisons (Appendix A). Furthermore, we analyzed the genes that were altered when U87-GSCs were treated with MRS1220 under normoxic conditions; of the 9360 genes identified, upregulation was observed in 2110 genes (22%), while 2224 (24%) genes were downregulated (Appendix A). For the validations of the RNAseq, we selected the *TRIB2*, NOTCH2, and NEF2L2 genes for two main reasons: first, because they have biological relevance (chemoresistance, hypoxia, and GSC phenotype) and because they are part of the set of genes with the most remarkable change induced by hypoxia. The *TRIB2* transcript analyzed by RTqPCR in U87-GSC was upregulated under hypoxic conditions, with an approximate 0.5-fold change (Appendix A). The NOTCH2 and NF2L2 transcripts were downregulated in U87-GSC under hypoxic conditions, with approximate changes of 0.7- and 0.4-fold, respectively (Appendix A). Furthermore, we validated the same genes in a GBM27 primary culture. The *TRIB2* transcript analyzed by RTqPCR in GBM was upregulated under hypoxic conditions, with an approximate 1.5-fold change (Appendix A). The NOTCH2 and NF2L2 transcripts were downregulated in U87-GSC under hypoxic conditions, with approximate changes of 0.35- and 0.4-fold, respectively (Appendix A). The three genes analyzed correlate with the results obtained in the RNAseq. These results indicate broad genetic dysregulation in cells cultured under hypoxic conditions versus those cultured under normoxic conditions with or without MRS1220 treatment. These results suggest that the switch from normoxia to hypoxia results in a more significant dysregulation of transcript expression than that observed using the A3AR antagonist MRS1220 alone.

### 2.2. Analysis of Differentially Expressed Genes in U87-GSCs under Hypoxic Conditions with and without MRS1220 Treatment

To better understand the impact of MRS1220 treatment on the expression of transcripts in the hypoxic condition, we performed a comparative analysis between both conditions. We showed the top genes regulated by MRS1220 under hypoxic conditions in a heat map, considering the upregulated and downregulated genes (Appendix A). Under hypoxic conditions, after administration of MRS1220 treatment, 10,118 genes were identified, of which only 64 were upregulated and 37 were downregulated (Figure 2A). Differentially regulated genes were obtained from the volcano plot formed in the comparison of hypoxia versus hypoxia with MRS1220 treatment (Appendix A). When comparing each condition, many deregulated genes were observed after MRS1220 treatment under normoxic conditions, whereas such dysregulation was not observed under hypoxic conditions. Under hypoxic conditions, MRS1220 treatment resulted in marginal genetic dysregulation (1%). Using a Venn diagram, we established the overlap between the genes that were positively regulated by a hypoxic versus a normoxic condition and the genes that decreased their expression with MRS1220 treatment in hypoxic conditions (Figure 2B). In this respect, treatment with MRS1220 reduced the transcripts of 29 genes that could tentatively favor the tumor phenotype; three targets that promote chemoresistance, *LIMD1*, *TRIB2*, and *TGFB1*, were distinguished. Furthermore, the molecular functions, biological processes, and pathways related to the down- and upregulated genes were analyzed after MRS1220 treatment under hypoxic conditions (Table 1, Appendix A).

### 2.3. LIMD1, TRIB2, and TGFB1 Are Possible Therapeutic Targets of MRS1220 That Correlate with the Expression of Factors Induced by Hypoxia and Ectonucleotidases Related to the Purinergic Pathway

We suggest that *LIMD1* is a target regulated by MRS1220 in GSCs under hypoxic conditions related to chemoresistance. For this reason, through analysis of brain tumor expression data in the GlioVis application, we determined the relevance of this marker as a therapeutic target associated with hypoxia, purinergic signaling, and the poor prognosis of the disease. Firstly, with the GlioVis CCGA data, we performed Kaplan–Meier estimator survival analyses with 633 cases of brain tumors (Figure 3A). Of these cases, 315 had high expression of *LIMD1*, and 318 had low expression of the marker. We detected that the median survival for cases with high expression of *LIMD1* was 33 months compared to 86.6 months for cases with low marker expression. Through a comparative analysis of the different histologies of brain tumors, we detected that *LIMD1* was significantly more regulated in GB than other brain tumors (Figure 3B). Subsequently, with data from patients with GB, we analyzed whether *LIMD1* correlated with factors induced by hypoxia, such as *HIF1A* (Figure 3C) and *HIF2A* (Figure 3D). We detected that the expression of *LIMD1* had a positive correlation with both factors, noting that the strongest positive correlation was with *HIF1A*. Likewise, we analyzed whether *LIMD1* correlated with the expression of the ectonucleotidases related to purinergic signaling of *NT5E* (CD73) and *ACP3* (PAP) (Figure 3E,F). We detected that *LIMD1* had a statistically significant positive correlation with both ectonucleotidase genes. We also analyzed the possible correlation of *LIMD1* with the A3AR (*ADORA3*), where we found no statistical association (Appendix A). These results indicate that *LIMD1* is a marker of poor prognosis in brain tumors and that it correlates with the expression of hypoxia-inducible factors (HIFs) and ectonucleotidases involved in purinergic signaling in GB. *LIMD1* makes it a potential target for the action of MRS1220 in GSCs in a hypoxic environment. With the same approach, we analyzed expression data in brain tumors in the GlioVis application; we determined the relevance of this marker as a therapeutic target associated with hypoxia, purinergic signaling, and the poor prognosis of the disease.

Regarding *TRIB2*, we performed a Kaplan–Meier estimator survival analysis with the GlioVis brain tumor cases (Figure 4A). Of these cases, 316 had high expression of *TRIB2*, and 317 had low expression of the marker. We found that the median survival for cases with high expression of *TRIB2* was 30.1 months compared to 88.6 months for cases with low marker expression. Through a comparative analysis of the different histologies of brain tumors, we detected that *TRIB2* had higher levels in anaplastic astrocytoma and GB than the other histologies (Figure 4B). Subsequently, using data from GB patients, we analyzed the correlation of *TRIB2* expression with HIFs, such as *HIF1A* (Figure 4C) and *HIF2A* (Figure 4D). In this respect, we detected that *TRIB2* had correlations with both factors, considering that the positive correlations were quite similar with *HIF1A* and *HIF2A*. Furthermore, we analyzed whether *TRIB2* correlates with the expression of the purinergic signaling-related ectonucleotidases *NT5E* (Figure 4E) and *ACP3* (Figure 4F). *TRIB2* had a statistically significant positive correlation with both ectonucleotidase genes. However, it had the highest positive correlation with the *NT5E* gene. We also analyzed the possible correlation of *TRIB2* with *ADORA3*, but we did not find that it maintained a statistically significant positive correlation (Appendix A). These results indicate that *TRIB2* is a marker of poor prognosis in brain tumors and correlates with the expression of hypoxia-inducible factors, ectonucleotidases, and the A3AR gene involved in purinergic signaling in GB. *TRIB2* is to be pointed out as a potential target for MRS1220 action on GSCs in a hypoxic environment.

Finally, we analyzed *TGFB1* as a potential gene associated with hypoxia, purinergic signaling, and the poor prognosis of the disease. Like the genes previously studied, we performed a survival analysis using the Kaplan–Meier estimator with the cases of GlioVis brain tumors (Figure 5A). Of these cases, 316 had high expression of *TGFB1*, and 317 had low expression of the marker. We found that the median survival for cases with high expression of *TGFB1* was 33.7 months compared with 98.8 months for cases with low marker expression. Through comparative analysis of different brain tumor histologies, we detected that *TGFB1* mRNA had the highest levels in astrocytoma, anaplastic astrocytoma, and GB compared to the other histologies (Figure 5B). Subsequently, using data from GB patients, we analyzed the correlations of *TGFB1* expression with HIFs, such as *HIF1A* (Figure 5C) and *HIF2A* (Figure 5D). In this respect, we detected that *TGFB1* had a statistically significant correlation with both factors, considering that the positive classification was quite solid and similar for *HIF1A* and *HIF2A*. Furthermore, we analyzed whether *TGFB1* correlated with the expression of the ectonucleotidases *NT5E* (Figure 5E) and *ACP3* (Figure 5F). In this respect, *TGFB1* had a high and positive score with both ectonucleotidase genes. We also analyzed the possible sorting of *TGFB1* with *ADORA3*, and we did not find that they maintained a statistically substantial high positive sorting (Appendix A). Finally, we can indicate that *TGFB1* is a marker of poor prognosis in brain tumors and correlates with the expression of HIFs, ectonucleotidases, and the *ADORA3* gene involved in purinergic signaling in GB. *LIMD1*, *TRIB2*, and *TGFB1* should be identified as potential targets for MRS1220 action on GSCs in a hypoxic environment.

## 3. Discussion

Different mechanisms of chemoresistance exist in glioblastoma, such as the activation of methylguanine methyltransferase (MGMT), increased activity of ATP-binding transporters, tumor heterogeneity, the immunosuppressive microenvironment, and GSCs [19]. Fighting GSCs seems to be a good strategy for treating glioblastoma. However, in the clinic, there are still no specific treatments for this type of cell population, and finding an appropriate therapy depends greatly on the individual molecular characteristics of the tumor [20]. Among the different strategies used are attacking the GSC niche, viral therapy with oncolytics, chimeric antigen receptor T (CAR T) cell therapy, targeting markers of GSCs, and inhibiting autophagy [21]. Treatment with temozolomide (TMZ) is the primary chemotherapy currently used for glioblastomas, but its effectiveness remains low, and its use is associated with a selective pressure that leads the tumor to develop chemoresistance against the drug [22,23]. Approaches with adjuvant therapies have improved the effects in preclinical trials in glioblastoma, and among them, the impact of blocking purinergic signaling and combined action with TMZ has been tested [23]. We previously demonstrated the relevance of treatment with MRS1220 in reducing blood vessel formation and tumor growth, and it would enhance its action with the data published in this report [24]. With the data found, a path is opened to detect a possible action as an adjuvant for TMZ therapy. 

In this article we demonstrated that A3AR antagonism using MRS1220 in GSCs under hypoxia regulates the expressions of *LIMD1*, *TRIB2*, and *TGFB1* genes, which are involved in chemoresistance and correlate with a poor prognosis, hypoxia, and purinergic signaling in GB. For GB, the tumor microenvironment is critical for the development and survival of GSCs [25]. The hypoxic microenvironment, a common hallmark of this type of cancer, is strongly linked to several mechanisms that contribute to radio- and chemoresistance, such as the expression of ATP-binding cassette (ABC) proteins [26]. Given the relevance of hypoxia in GB, we performed a targeted investigation of cells cultured under hypoxic conditions (with and without MRS1220 treatment). This choice was further justified by evidence demonstrating that hypoxia increases the expression of stem cell markers and promotes clonogenicity in GB neurospheres [27]. Hypoxic niches are vital for regulating GSC activities such as proliferation, self-renewal, and maintenance of potency [28]. Li et al. (2009) first reported the effects of hypoxia and low oxygen levels on GSCs. GSCs preferentially express higher levels of the HIF family of transcription factors and multiple HIF-regulated genes compared to non-stem tumor cells and normal neural progenitors [29]. HIF dimer positively regulates the transcription of downstream genes involved in cell survival, motility, metabolism, and angiogenesis and contributes to the potency and self-renewal of GSCs [30]. In vitro assays revealed that the migration and invasion of GSC lines were markedly increased under severely hypoxic conditions of 0.5–1% O_2_ compared to normoxic conditions [31,32]. Hypoxic GB environments have a higher level of CBF1, an essential transcriptional regulator of Notch signaling, which contributes to maintaining GSC and regulating epithelial–mesenchymal transition (EMT) [33]. Hypoxia prompts the anaerobic glycolysis pathway to generate energy in tumor cells in combination with the production of acidic metabolites, mainly including lactic acid. For this reason, hypoxia reduces the pH in solid tumors, promoting the GSC phenotype [34]. A mean partial pressure of oxygen (pO2) of 5 to 9 mm Hg and an acidic pH of 6.8 or less often exists in tumors from GB patients and in murine GB xenograft models [35,36]. However, oxygen tension varies significantly across non-neoplastic tissue, indicating the importance of establishing physiological control of normoxia when designing experiments to uncover differences in these microenvironmental conditions [34]. In a hypoxic microenvironment, the HIF1α/HIF2α-Sox2 network induced the formation of GSCs by dedifferentiating differentiated glioma cells, thereby promoting chemoresistance of the glioma cells [37]. Regarding this last point, the induction of multidrug resistance (MDR) is observed during hypoxia through the expression and activity of ABC transporters [38,39]. In this context, it has been proposed that ADO, a nucleoside that increases in the extracellular space under hypoxic conditions, plays a fundamental role in the induction of MDR mechanisms [40]. It has been proposed that ADO-activated signaling through its receptors is crucial for acquiring malignant tumor characteristics, such as cell migration and invasion. Our group previously demonstrated that activated A_3_ receptor signaling increases these malignant characteristics, especially in GSCs [32]. Furthermore, we have previously pointed out that A_3_ antagonism can generate a chemosensitizing effect in GB cells and that this was partly due to the regulation of the MRP1 transporter [17]. Despite these findings, little has been elucidated about the mechanisms by which A3AR antagonism could favor cell chemosensitization. Through RNAseq and RTqPCR validations, we observed that NOTCH2 and NFE2L2 were downregulated in hypoxic conditions when we compared them to the normoxic conditions. Firstly, in the context of endothelial cells, it has been reported that the downregulation of NOTCH2 in a hypoxic environment can promote pro-tumor properties, such as migration or proliferation, mainly guided by the increase of NOTCH1 (as a compensatory mechanism for the decrease in NOTCH2) that is related to GSCs, as well as glioblastoma malignancy [41,42]. Regarding the regulation of NFE2L2, it has been observed that its expression can be negatively regulated under hypoxic conditions (PMID: 38424190). However, NFE2L2 signaling is intricate and may be accompanied by different changes in HIF1a expression that play a critical role in hypoxic conditions [43]. With the findings of this study, we detected three possible candidate genes associated with chemoresistance to different drugs in cancer.

Regarding *LIMD1*, it has been identified in colorectal cancer that its expression correlates with MDR, and it has also been established that its negative regulation can reverse drug resistance in multidrug-resistant cells of the same type of cancer [44]. Foxler D. et al. determined that *LIMD1* is a hypoxia-induced gene that regulates the degradation of HIF1-α via the proteasome [45]. Specifically, the *LIMD1* gene responds to hypoxia through an HIF1-α response element in its promoter and, as a protein, acts as a scaffold to stabilize the PHD2/VHL/HIF-1α degradation complex, in this sense working as a tumor suppressor [46]. Furthermore, it has been reported that *LIMD1* regulates the cell cycle and entry into the synthesis phase through a mechanism dependent on retinoblastoma protein (Rb) and E2F [47]. The function of *LIMD1* has not yet been characterized in brain tumors, but we detected that overexpression of this marker is associated with a poor prognosis of the disease. Furthermore, we observed a correlation between markers induced by hypoxia and *LIMD1*, which correlates with previous reports [45]. Interestingly, we detected that the expression of *LIMD1* correlates with the expression of ectonucleotidases that participate in purinergic signaling, and through RNAseq analysis, we observed that A3AR antagonism decreases the expression of this marker in a GSC model in a hypoxic environment. There is no apparent connection between signaling induced by the A3AR and *LIMD1* in the literature, which may give rise to future research in the area.

Regarding *TRIB2*, it can function as an oncogene, regulating a wide range of cellular processes, including tumorigenesis, proliferation, invasion, and chemoresistance in various cancer subtypes [48,49,50]. Silencing *TRIB2* has been observed to reduce cell proliferation, colony formation, and wound healing in melanoma cells. In an in vivo melanoma xenograft model, *TRIB2* knockdown significantly reduced tumor growth [51]. *TRIB2* has been reported to interact with MAPKK, AKT, and NFkB proteins, playing a role in cell survival, proliferation, and immune response [52,53]. Inhibition of *TRIB2* has been shown to resensitize resistant prostate cancer cells to enzalutamide [54]. In GB, *TRIB2* is correlated with a pathological classification, radioresistance, and temozolomide (TMZ) resistance [55]. In this study, we associate the elevation of *TRIB2* in GSCs with hypoxia and HIF proteins. Moreover, we observed decreased *TRIB2* expression in GSCs cultured in hypoxia and treated with MRS1220. This finding suggests the A3AR as a potential therapeutic target in GB. The evidence linking the blockade of the A3AR to *TRIB2* expression and chemoresistance provides an attractive opportunity for further studies on chemoresistance.

Finally, in this study, we demonstrated that the TGB1 transcript increased its expression in GSCs under hypoxia and observed that this expression decreased upon in vitro treatment of MRS1220. TGF-β is secreted by gliomas, and its expression increases under hypoxia due to the activity of NANOG, a transcription factor induced by HIF-1α, which promotes the stemness signature of GSCs [56,57]. The maintenance of the GSC phenotype appears to be supported by TGF-β, which engages an autocrine feedback loop that maintains stemness via increased expression of the sex-determining region Y box (Sox) 2 and Sox4 [58]. It has been reported that ADO can regulate the expression and secretion of TGF-β [59,60]. However, this is the first time that A3AR blockades have been shown to regulate *TGFB1* expression in GSCs under hypoxia.

In patients with malignant glioma, it was found that levels of TGF-β protein were high in the blood serum and the tumor tissue; at the same time, these levels were correlated with the malignancy of the tumor, the stage of tumor development, and patient prognosis [61,62]. Similarly, our analysis demonstrated that high *TGFB1* expression in GB correlates with a poor prognosis. One term addressed is the so-called “TGF-β paradox”, which refers to the dual impact of TGF-β on cancer progression, as TGF-β can act as a potent tumor suppressor in early-stage tumors by inducing robust antiproliferative responses, cellular differentiation, and apoptosis [63]. However, in advanced-stage cancer, TGF-β has the potential to promote many of the malignant features of GB, such as migration/invasion, angiogenesis, immunosuppression, and drug resistance/radioresistance [63,64,65,66]. Specifically, in our analysis, we detected that high expression of *TGFB1* in GB correlates with a poor prognosis. Bruna et al. found that high TGF-β/Smad activity confers a poor prognosis on glioma patients and promotes cell proliferation through platelet-derived growth factor B (PDGF-B) [67,68]. TGF-β has previously been reported to induce resistance to chemotherapy, targeted therapy, and immunotherapy [69]. In GBM, TGF-β contributes to TMZ resistance by increasing MGMT accumulation and repressing miR-198 levels [70]. Furthermore, it has been detected that the same treatment with TMZ increases the expression of *TGFB1* [70]. Our bioinformatic analysis demonstrated that high expression of *TGFB1* in GB correlates with the expression of the extracellular ADO axis ectonucleotidases *NT5E* and *ACP3*. The relationship between the TGF-β pathway and the extracellular ADO axis has been reported in different disease models [59,60,71,72,73,74,75]. TGF-β/ADO signaling generates a feedback loop because extracellular ADO induces the production and secretion of TGF-β, which maintains CD73 expression [71,72,76]. 

In conclusion, our results suggest that A3AR antagonism can negatively regulate the expression of markers related to chemoresistance, such as *LIMD1*, *TRIB2*, and *TGFB1*, in GSCs in hypoxia, which may bring advantages to therapies for GB. However, a limitation is the availability of antagonists of the A3AR that have been proven in humans and blockers of the pathways involving *LIMD1*, *TRIB2*, or *TGFB1* (Figure 6). Additionally, the consequences of A3AR blockades on physiological functions are still unknown, including how they affect their role in increasing ischemic/hypoxic tolerance to cardiomyocytes, thereby reducing cardiovascular damage and infarcts [77]. In this regard, a delivery system that could help direct drug delivery to brain tumors may be preferred.

## 4. Materials and Methods

### 4.1. Cell Line Culture and Primary Culture

The human GB U87MG cell line was acquired from the American Type Culture Collection Company (ATCC^®^ HTB-14TM., Manassas, VA, USA). The GBM27 primary culture from a GB was kindly donated by Dr. Ángel Ayuso-Sacido [78]. The U87MG cells were grown in T-75 culture flasks until they reached 70 to 80% confluence. D-MEM-F12 medium supplemented with 10% fetal bovine serum (FBS) and 1% penicillin/streptomycin was used. The GBM27 culture was grown in M21 medium, which consists of DMEM/F12, non-essential amino acids (10 mM), Hepes (1 M), D-glucose (45% G8769), BSA-F5 (7.5%), pyruvate sodium (100 mM), L-Glutamine (200 mM), antibiotic-antimycotic (100×), N_2_ supplement (100×), hydrocortisone (1 ug/uL), triiodothyronine (100 ug/mL), EGF (25 ng/uL), bFGF (25 ng/uL), and Heparin (1 ug/uL). Culture was carried out under a controlled atmosphere of 5% CO_2_, a relative humidity of 95%, and a temperature of 37 °C. For the transfer of the cells, the culture medium was removed by aspiration, and the cell monolayer was washed twice with PBS 1X. It was then incubated with 1 mL of 0.05% trypsin in 0.1% EDTA for 5 min at 37 °C. Subsequently, trypsin was neutralized with 1 mL of complete culture medium, reaching a trypsin-and-culture medium ratio of 1:1. The cells were recovered by centrifugation at 600× *g* for 5 min, and the cell pellet was resuspended in fresh medium at the required density according to the procedure to be followed, and later they were seeded in new T-75 culture flasks.

### 4.2. GB Stem-like Cell (GSC) Culture

Once the U87MG cells were cultured, they were seeded in 6-well plates with neurobasal medium (GIBCO). These media were supplemented with 20 ng/mL EGF, 20 ng/mL bFGF, B27 (1×), Glutamax 1×, 1% penicillin/streptomycin, and 2 mM L-glutamine. The culture was carried out under standard conditions at 5% CO_2_ (normoxic condition) and 0.5% O_2_ (hypoxic condition), at 37 °C and 95% relative humidity in both cases. After 7 days of culture, the treatment with MRS1220 was applied. 

### 4.3. Pharmacological Agents

For the normoxic condition, U87MG-GSC cells were treated directly with MRS1220 (10 mM) as a selective antagonist of the A3AR [17] or with 0.001% DMSO as a vehicle for 24 h to subsequently collect the material. For the hypoxic condition, U87MG-GSC cells were incubated for 24 h with MRS1220 (10 mM) or with 0.001% DMSO in a controlled hypoxia chamber that maintained O_2_ levels at 0.5%.

### 4.4. RNA Isolation and Sequencing

RNA was extracted using “NucleoSpin^®^ RNA” according to the manufacturer’s instructions (Macherey-Nagel, Inc., Düren, Germany). The quality of total RNA isolated from U87MG-GSCs treated with vehicle and MRS1220 under standard culture conditions was measured with the Fragment Analyzer (Advanced Analytical Technologies, Ankeny, IA, USA), considering an RNA Quality Number (RQN) equal or superior to 8 for library preparation. The RNA-seq library was performed using the TruSeq RNA Sample Preparation Kit (Illumina, Inc., San Diego, CA, USA), and its quantitation was performed by qPCR using the Library Quant Kit Illumina GA (KAPA), following the manufacturer’s instructions. The generated libraries were clustered on-board and sequenced to generate 125b PE reads using the high-throughput sequencing system HiSeq2500 (Illumina, Inc., San Diego, CA, USA). The sequences were mapped to the HS_GRCh38 human genome (ensembl.org), and the number of read counts per gene was determined for each library using the feature counts function of the Rsubread R library. To determine differential expression based on raw counts, we used the DEseq2 R library, and an adjusted *p*-value equal to or less than 0.05 was considered statistically significant. Eight transcriptomes were analyzed: two from U87MG-GSCs treated with vehicle, two from U87MG-GSCs treated with MRS1220, two from U87MG-GSCs in hypoxic conditions treated with vehicle, and two from U87MG-GSCs in hypoxic conditions treated with MRS1220.

### 4.5. Functional Genomic Analysis of RNAseq Data

Different analytical approaches were applied for functional genomics: DAVID (Database for Annotation, Visualization, and Integrated Discovery) (https://david-d.ncifcrf.gov/, accessed on 13 November 2023) provides a comprehensive set of functional annotation tools for investigators to understand the biological meaning behind large lists of genes. The Venn diagram (http://bioinformatics.psb.ugent.be/webtools/Venn/, accessed on 15 November 2023) tool was used to calculate the intersection(s) of list of elements. Ensemble (https://www.ensembl.org/index.html, accessed on 15 November 2023), a genome browser for vertebrate genomes, was used to predict regulatory functions. Panther (http://www.pantherdb.org/, accessed on 15 November 2023) provides comprehensive information about the evolution of protein-coding gene families, particularly protein phylogeny, functions, and genetic variation impacting those functions. INSECT 2.0 (IN-silico SEarch for Co-occurring Transcription factors) (http://bioinformatics.ibioba-mpsp-conicet.gov.ar/INSECT2/, accessed on 15 November 2023) is a web server for biologists analyzing genomic sequence data for in silico cis-regulatory module prediction and analysis. GlioVis (http://gliovis.bioinfo.cnio.es/, accessed on 19 December 2023) was used as the data visualization tool for brain tumor datasets.

### 4.6. RT-qPCR

To validate the RNAseq results, we performed RNA extraction with TRIzol RNA Isolation Reagents (Invitrogen, Waltham, MA, USA), and subsequently NanoDrop quantified the samples. Reverse transcription was performed with 1 μg of RNA with MMLV (Promega, Madison, WI, USA) according to the manufacturer’s instructions. The relative quantification of RT-qPCR was performed with the 2^−∆∆Ct^ method, and the β-actin gene was used as a normalizer. For each RT-qPCR reaction, 12.5 µL of Brilliant II SYBR Green QPCR master mix (Agilent Technologies, Santa Clara, CA, USA), 0.5 µM of forward primer, 0.5 µM of reverse primer, nuclease-free water, and 1 µL of template were added to cDNA. The sequences of the primers used are as follows: *TRIB2* forward 5′GTTTTTCGTGCCGTGCATCT3′, *TRIB2* reverse 5′GTCCCCATAGCTTCGCTCAA3′, NOTCH forward 5′TGAAGTGGATGAGTGCCAGAA3′, NOTCH reverse 5′CCATGCACTGACCACCATTAAG3′, NFE2L2 forward 5′AGGTTGCCCACATTCCCAAA3′, NFE2L2 reverse 5′AGTGACTGAAACGTAGCCGA3′, β-actin forward 5′GAGCACAGAGCCTCGCCTTT3′, and β-actin reverse 5′CACGATGGAGGGGAAGACG3′. All data were recorded in biologic triplicate.

### 4.7. Statistical Analysis

Comparisons between multiple groups were performed using one-way ANOVA and Tukey’s post hoc test. For Pearson correlations, analyses were performed to look at the data distribution. In addition, it was adjusted to a 95% confidence interval. All statistical tests were performed as two-sided and were considered significant at a *p*-value <0.05. Statistical analyses were performed using GraphPad Prism 8 software (GraphPad Software, Inc., San Diego, CA, USA).

## Figures and Tables

**Figure 1 pharmaceuticals-17-00579-f001:**
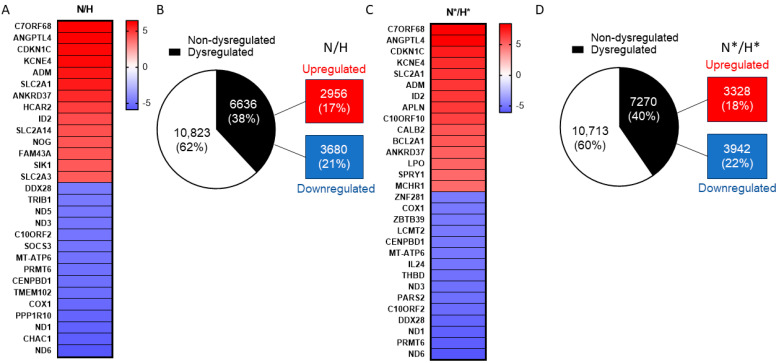
Effect of MRS1220 on GSCs cultured under normoxic versus hypoxic conditions. (**A**) A heat map of the genes altered in GSC under normoxic versus hypoxic conditions, with the genes with the most significant positive regulation in red and the genes with the most negative regulation in blue, considering the fold change. (**B**) Circular graph showing the number and percentage of deregulated and non-deregulated genes in hypoxic versus normoxic conditions (without MRS1220 treatment); the red box indicates the number and percentage of genes upregulated in that condition, and the blue box indicates the number and percentage of genes downregulated in that condition. (**C**) A heat map of the altered genes in GSC under normoxic versus hypoxic conditions with treatment with 10 mM MRS1220, with the genes with the most significant positive regulation in red and the genes with the greatest negative regulation in blue, considering the fold change. (**D**) Circular graph showing the number and percentage of deregulated and non-deregulated genes in normoxic versus hypoxic conditions with 10 mM MRS1220 treatment; the red box indicates the number and percentage of upregulated genes in that condition, and the blue indicates the number and percentage of genes downregulated in that condition. N = normoxia; H = hypoxia; * = MRS1220.

**Figure 2 pharmaceuticals-17-00579-f002:**
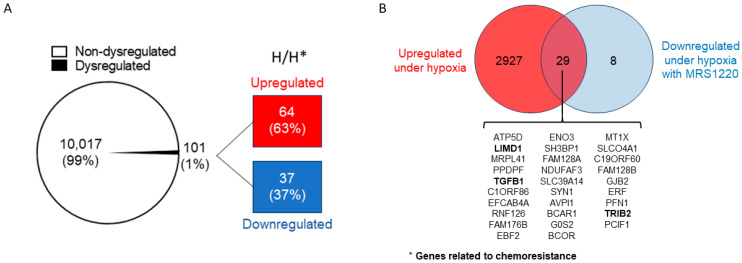
Differentially expressed genes in GSCs after A3AR blockades using MRS1220 in a hypoxic environment. (**A**) Circular graph showing the numbers and percentages of dysregulated and non-dysregulated genes under hypoxic conditions versus hypoxia with MRS1220 treatment; red box indicates the number and percentage of upregulated genes, and blue box indicates the number and percentage of genes downregulated by MRS1220 in that condition. (**B**) The Venn diagram shows the genes hypoxia regulates in the red circle, and the blue circle shows differentially regulated genes that were downregulated in the hypoxic condition when treated with MRS1220. The overlap shows the genes that were regulated in both conditions; at the bottom, the list of the names of the altered genes is shown, and those that are directly related to chemoresistance are in bold. H = hypoxia; * = MRS1220.

**Figure 3 pharmaceuticals-17-00579-f003:**
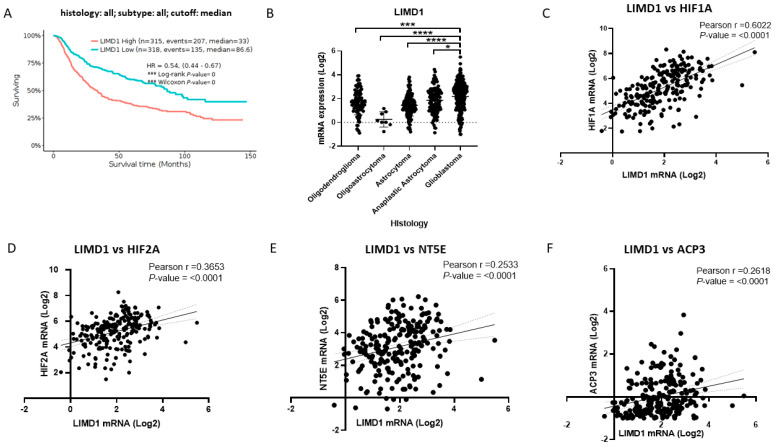
*LIMD1* is a potential therapeutic target of MRS1220 in GB, and it positively correlates with the expression of hypoxia-inducible factors and ectonucleotidases. (**A**) Survival analysis with Kaplan–Meier graph of 633 cases of brain tumors (anaplastic oligodendroglioma (N = 58), oligodendroglioma (N = 91), oligoastrocytoma (N = 8), anaplastic astrocytoma (N = 116), anaplastic oligoastrocytoma (N = 15), astrocytoma (N = 124), and glioblastoma (N = 221)); the red line represents tumors with high levels of *LIMD1*, and the blue line represents tumors with low levels of *LIMD1*; the graph reports the median survival and Hazard Ratio (HR) in conjunction with statistical significance. (**B**) Analysis of *LIMD1* mRNA expression in different brain tumor histologies. (**C**) Pearson correlation of *LIMD1* mRNA expression with *HIF1A*. (**D**) Pearson correlation of *LIMD1* mRNA expression with *HIF2A*. (**E**) Pearson correlation of *LIMD1* mRNA expression with *NT5E*. (**F**) Pearson correlation of *LIMD1* mRNA expression with *ACP3*. Pearson r- and *p*-values are shown in all correlation graphs. The data were obtained from GlioVis CCGA. Data are presented as the mean ± standard deviation (SD); * *p* < 0.05; *** *p* < 0.001; and **** *p* < 0.0001.

**Figure 4 pharmaceuticals-17-00579-f004:**
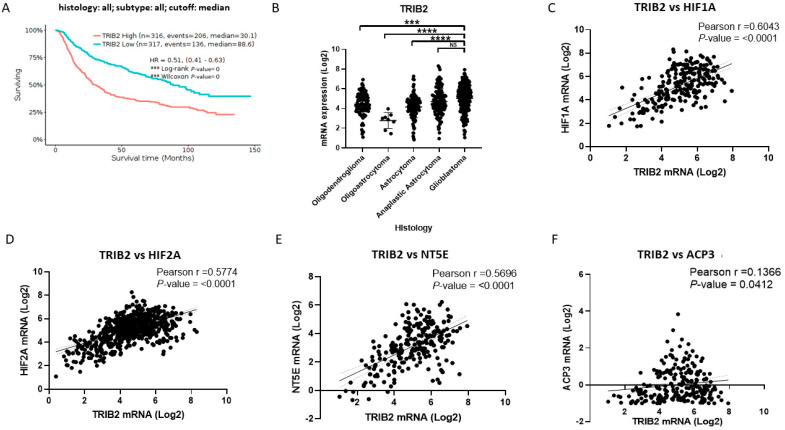
*TRIB2* is a potential therapeutic target of MRS1220 in GB, and it positively correlates with the expression of hypoxia-inducible factors and ectonucleotidases. (**A**) Survival analysis with Kaplan–Meier graph of 633 cases of brain tumors (anaplastic oligodendroglioma (N = 58), oligodendroglioma (N = 91), oligoastrocytoma (N = 8), anaplastic astrocytoma (N = 116), anaplastic oligoastrocytoma (N = 15), astrocytoma (N = 124), and glioblastoma (N = 221)); the red line represents tumors with high levels of *TRIB2*, and the blue line represents tumors with low levels of *TRIB2*; the graph reports the median survival and Hazard Ratio (HR) in conjunction with statistical significance. (**B**) Analysis of *TRIB2* mRNA expression in different brain tumor histologies. (**C**) Pearson correlation of *TRIB2* mRNA expression with *HIF1A*. (**D**) Pearson correlation of *TRIB2* mRNA expression with *HIF2A*. (**E**) Pearson correlation of *TRIB2* mRNA expression with *NT5E*. (**F**) Pearson correlation of *TRIB2* mRNA expression with *ACP3*. Pearson r- and *p*-values are shown in all correlation graphs. The data were obtained from GlioVis CCGA. Data are presented as the mean ± standard deviation (SD); NS, not significant; *** *p* < 0.001; and **** *p* < 0.0001.

**Figure 5 pharmaceuticals-17-00579-f005:**
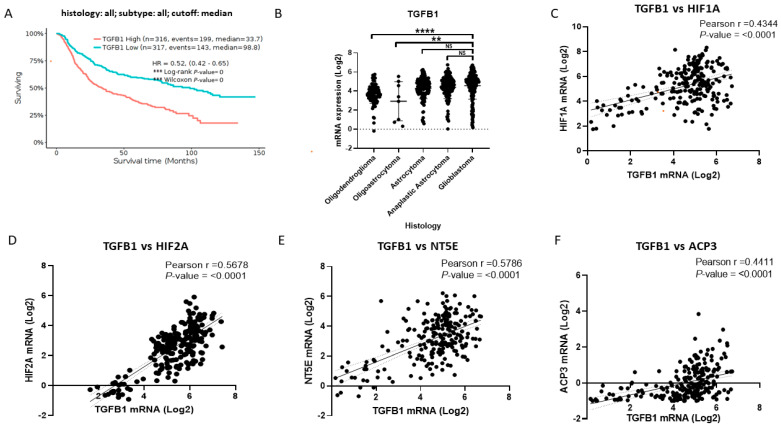
*TGFB1* is a potential therapeutic target of MRS1220 in GB, and it positively correlates with the expression of hypoxia-inducible factors and ectonucleotidases. (**A**) Survival analysis with Kaplan–Meier graph of 633 cases of brain tumors (anaplastic oligodendroglioma (N = 58), oligodendroglioma (N = 91), oligoastrocytoma (N = 8), anaplastic astrocytoma (N = 116), anaplastic oligoastrocytoma (N = 15), astrocytoma (N = 124), and glioblastoma (N =221)); the red line represents tumors with high levels of *TGFB1*, and the blue line represents tumors with low levels of *TGFB1*; the graph reports the median survival and Hazard Ratio (HR) in conjunction with statistical significance. (**B**) Analysis of *TGFB1* mRNA expression in different brain tumor histologies. (**C**) Pearson correlation of *TGFB1* mRNA expression with *HIF1A*. (**D**) Pearson correlation of *TGFB1* mRNA expression with *HIF2A*. (**E**) Pearson correlation of *TGFB1* mRNA expression with *NT5E*. (**F**) Pearson correlation of *TGFB1* mRNA expression with *ACP3*. Pearson r- and *p*-values are shown in all correlation graphs. The data were obtained from GlioVis CCGA. Data are presented as the mean ± standard deviation (SD); NS, not significant; ** *p* < 0.01 and **** *p* < 0.0001.

**Figure 6 pharmaceuticals-17-00579-f006:**
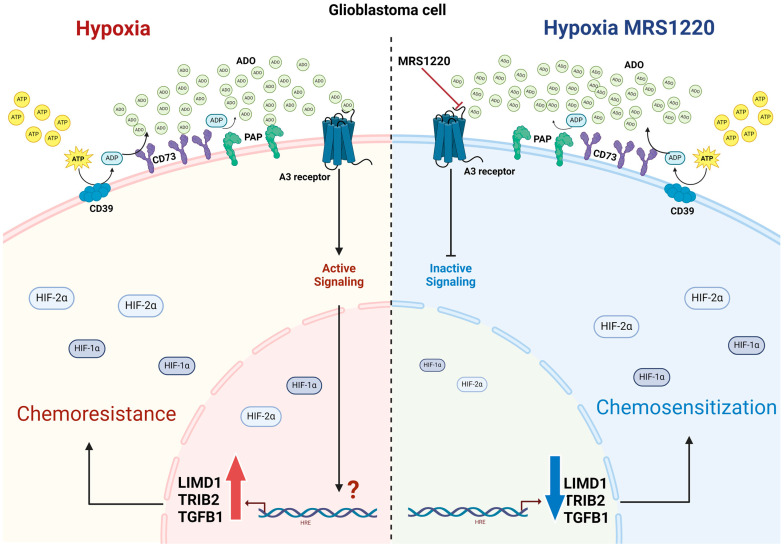
Proposed model made with Biorender (accessed 21 April 2024). red arrow; up-regulation, blue arrow; down-regulation.

**Table 1 pharmaceuticals-17-00579-t001:** Molecular functions, biological processes, and pathways related to the down- and upregulated genes after MRS1220 treatment in hypoxic conditions.

	Molecular Functions	Biological Processes	Pathways
29 downregulated genes	Binding Catalytic activity Molecular function regulator Structural molecule activity Transcription regulator activity Transporter activity	Biological Regulation Cellular process Developmental Process Localization Locomotion Metabolic process Response to stimulus Signaling	CCKR signaling map Cytoskeletal regulation by Rho GTPase Integrin signaling pathway PDGF signaling pathway Synaptic vesicle trafficking
34 upregulated genes	ATP-dependent activity Binding Catalytic activity Molecular adaptor activity Molecular function regulator Molecular transducer activity Structural molecule activity Transcription regulator activity	Biological process involved in interspecies interaction between organisms Biological regulation Cellular process Developmental process Localization Locomotion Metabolic process Multicellular organismal process Response to stimulus Signaling	Alzheimer disease: presenilin pathway Angiogenesis EGF receptor signaling pathway FGF signaling pathway Gonadotropin-releasing hormone receptor pathway Inflammation mediated by chemokine and cytokine signaling pathway Notch signaling pathway PDGF signaling pathway

## Data Availability

Data are available on the GlioVis (http://gliovis.bioinfo.cnio.es/) accessed on 3 January 2024. data visualization tool for brain tumor datasets.

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
