# Peer review of "The Impact of A3AR Antagonism on the Differential Expression of Chemoresistance-Related Genes in Glioblastoma Stem-like Cells"

_pharmaceuticals, 2024, doi:10.3390/ph17050579_

Round 1
Reviewer 1 Report
Comments and Suggestions for Authors
In the following study authors analyzed genes affected by A3AR blockade under normoxic and hypoxic conditions, identifying potential candidates like LIMD1, TRIB2, and TGFB1, associated with chemoresistance and regulated by hypoxia and A3AR blockade in glioblastoma. Hypoxic conditions induce differential gene expression in GSCs, increasing chemoresistance-associated genes. MRS1220, an A3AR antagonist, can regulate expression of these genes, linked to poor prognosis, hypoxia, and purinergic signaling.
This work doesn’t reach a cconsitnet level to be publish on this journal and I am force to reject it.
1) The experiments are conducted on one single cell line which is not enough.
2) There is no report of the number of transcriptomes performed and analyzed to obtained the data.
3) Figure 1 and Figure 2 are not well described. For example in Figure 1B is not clear what the authors are desribed. The descritoion in the legend is not clear.
4) RT-PCR analyses and results are mention bu there is no figure associated to it.
5) There is no clear explanation of the glioblastoma datasets used for the anlyses. In figure 3 authors report 633 brain tumors. It is not clear what kind of brain tumors. In material and methods the authors mention the use of the database GLIOVIS, but in GLIOVIS there are no, as far as I know there is not a glioblastoma dataset with such a high numer of samples.
6) The conclusions with data presented in such a way are not acceptable
The study under review explores the impact of A3AR blockade on gene expression under both normoxic and hypoxic conditions, with a focus on identifying potential candidates associated with chemoresistance in glioblastoma. The authors highlight genes such as LIMD1, TRIB2, and TGFB1, which are regulated by hypoxia and A3AR blockade.
However, several significant shortcomings preclude its suitability for publication:
1) Limited Cell Line Representation: The study primarily relies on experimentation with a single cell line, which undermines the generalizability of the findings. Multiple cell lines should be used to establish the robustness and broader applicability of the results.
2) The number of transcriptomes conducted and analyzed to obtain the data is not provided. This lack of transparency raises concerns about the reproducibility and reliability of the results.
3) Inadequate Figure Descriptions: Figures 1 and 2 lack clear descriptions, hampering the understanding of key findings. For instance, Figure 1B requires clarification regarding the information presented, as the legend fails to provide sufficient detail.
4) Missing RT-PCR Figures: Although RT-PCR analyses are mentioned, it is not reported where to look for the results. However the supplementary figures contain the graph of the RT PCR, but again no mentioning of details of the procedure are reported.
5) Unclear Description of Datasets: The explanation of the glioblastoma datasets used for analyses is ambiguous. For example, the mention of 633 brain tumors in Figure 3 lacks specificity regarding the types of brain tumors studied. Furthermore, discrepancies exist regarding the utilization of the GLIOVIS database, as it does not contain a glioblastoma dataset with such a large sample size.
6) Therre is overlapping between studyin hypoxia vs normoxic condition and MRS1220 treatment in hypoxic and mormoxic conditions. It is not clear how thay can distinguish the two different situations and what it is due to treatment and what to
Inadequate Conclusions: The conclusions drawn from the presented data lack coherence and substantiation. Without clearer presentation and analysis of the results, the validity of the conclusions is compromised.
In summary, while the study addresses an important topic, its methodological and reporting deficiencies prevent it from reaching the standard required for publication in this journal. Addressing these concerns through additional experimentation, improved figure descriptions, and enhanced clarity in data presentation is essential for further consideration.
Author Response
We greatly appreciate the comments provided. We make the suggested modifications and add the answers point by point.
- The experiments are conducted on one single cell line which is not enough.
R: We appreciate this review; we carried out new validations on GSCs from primary cultures exposed to hypoxia condition for strengthen our research. Figure 1 supplementary
- There is no report of the number of transcriptomes performed and analyzed to obtained the data.
R: Thank you very much for this observation, we added the necessary information in the materials and methods section on line 484.
- Figure 1 and Figure 2 are not well described. For example in Figure 1B is not clear what the authors are desribed. The descritoion in the legend is not clear.
R: We correct the description of the legends in both figures.
- RT-PCR analyses and results are mention bu there is no figure associated to it.
R: We added greater details of the RT-PCR results, description in materials and methods, and in the legend of the complementary figure.
- There is no clear explanation of the glioblastoma datasets used for the anlyses. In figure 3 authors report 633 brain tumors. It is not clear what kind of brain tumors. In material and methods the authors mention the use of the database GLIOVIS, but in GLIOVIS there are no, as far as I know there is not a glioblastoma dataset with such a high numer of samples.
R: We appreciate this valuable comment. Indeed, the survival analysis was carried out with the following brain tumors: Anaplastic oligodendrolgioma (N=58), oligodendroglioma (N=91), oligoastrocytoma (N=8), anaplastic atrocytoma (N=116), anaplastic oligoastrocytoma (N=15), astrocytoma (N=124) and glioblastoma (N = 221) this was clarified in the legend of Figures 3, 4, and 5.
- The conclusions with data presented in such a way are not acceptable
R: We added all the changes suggested by the reviewers and focused the conclusions on the changes in three markers of relevance to chemotherapy. We understand that our article has limitations, which we reflect in the article's discussion. In part, these limitations are due to the lack of clinical trials with this antagonist, which is why it is crucial to investigate the impact of the use of these drugs in different experimental situations. Currently, an A3 antagonist (PBF-1650) from the company Palobiofarma is in clinical trials to combat atopic dermatitis.
The study under review explores the impact of A3AR blockade on gene expression under both normoxic and hypoxic conditions, with a focus on identifying potential candidates associated with chemoresistance in glioblastoma. The authors highlight genes such as LIMD1, TRIB2, and TGFB1, which are regulated by hypoxia and A3AR blockade.
However, several significant shortcomings preclude its suitability for publication:
- Limited Cell Line Representation: The study primarily relies on experimentation with a single cell line, which undermines the generalizability of the findings. Multiple cell lines should be used to establish the robustness and broader applicability of the results.
R: We understand this concern, so we added validations in other GSC cell models to increase the study's robustness. Figure 1 supplementary
- The number of transcriptomes conducted and analyzed to obtain the data is not provided. This lack of transparency raises concerns about the reproducibility and reliability of the results.
R: We apologize for forgetting to add this valuable information. We add this information to materials and methods.
- Inadequate Figure Descriptions: Figures 1 and 2 lack clear descriptions, hampering the understanding of key findings. For instance, Figure 1B requires clarification regarding the information presented, as the legend fails to provide sufficient detail.
R: We made major changes to the figures description.
- Missing RT-PCR Figures: Although RT-PCR analyses are mentioned, it is not reported where to look for the results. However the supplementary figures contain the graph of the RT PCR, but again no mentioning of details of the procedure are reported.
R: We added more information and details about the RT-PCRs in the results, figure legends, and materials and methods section.
- Unclear Description of Datasets: The explanation of the glioblastoma datasets used for analyses is ambiguous. For example, the mention of 633 brain tumors in Figure 3 lacks specificity regarding the types of brain tumors studied. Furthermore, discrepancies exist regarding the utilization of the GLIOVIS database, as it does not contain a glioblastoma dataset with such a large sample size.
R: We add a detailed description of the analyzed samples in the figure legends.
- Therre is overlapping between studyin hypoxia vs normoxic condition and MRS1220 treatment in hypoxic and mormoxic conditions. It is not clear how thay can distinguish the two different situations and what it is due to treatment and what to
R: In the materials and methods section, we added a better description of the MRS1220 treatment procedure under hypoxic and normoxic conditions. Further, for each condition a transcriptomic analysis was performed allowing obtain differentially expressed mRNAs.

Reviewer 2 Report
Comments and Suggestions for Authors
The authors of the study used bioinformatics to prove the results of their earlier experimental work on GSCs. They found that when GSCs are located in hypoxic tumor niches, they generate high levels of extracellular adenosine. This causes the activation of the A3 adenosine receptor (A3AR), leading to chemoresistance. The authors then inhibited A3AR under hypoxic conditions in U87-GSCs. Using RNA seq and bioinformatics analyses, they discovered that MRS1220 can regulate the expression of LIMD1, TRIB2, and TGFB1 genes, which are responsible for chemoresistance. They also found that high expression of these genes correlates with poorer survival of glioblastoma patients by using GlioVis brain tumor cases.
While targeting A3AR components is a promising alternative for treating glioblastoma, the authors should discuss certain limitations of their study.
1. There is a possibility that targeting A3AR components may have unintended consequences on other cellular processes.
2. What are the possible alternative therapies for treating glioblastoma apart from targeting the components of GSCs?
3. There is a possibility that other factors besides GSCs contribute to chemoresistance in glioblastoma.
The authors should provide a detailed legend of Figure 6.
Author Response
We greatly appreciate the comments provided. We make the suggested modifications and add the answers point by point.
- There is a possibility that targeting A3AR components may have unintended consequences on other cellular processes.
R: Thank you very much for the valuable comment, we added this point to the discussion. Line 426
- What are the possible alternative therapies for treating glioblastoma apart from targeting the components of GSCs?
R: We add to the discussion different therapies that can be used for glioblastoma and GSCs. Line 487
- There is a possibility that other factors besides GSCs contribute to chemoresistance in glioblastoma.
R: Indeed, there are other chemoresistance mechanisms apart from GSCs, and we added a paragraph to the discussion mentioning these mechanisms. Line 481

Reviewer 3 Report
Comments and Suggestions for Authors
The paper focuses on the impact of A3 adenosine receptor (A3AR) antagonism on the differential expression of chemoresistance-related genes in glioblastoma stem-like cells. It presents a detailed analysis of how blocking A3AR affects the expression of certain genes under both normoxic and hypoxic conditions, identifying potential therapeutic targets for glioblastoma (GB), which is known for its poor prognosis and resistance to treatment. The study identifies three genes (LIMD1, TRIB2, and TGFB1) that could be regulated by MRS1220, an A3AR antagonist, under hypoxic conditions, suggesting a new avenue for GB treatment strategies.
Some comments need to be addressed.
1. additional validation in other GB cell lines are needed.
2. The paragraph 2.1 briefly mentions the outcome of the treatment (e.g., upregulation of TRIB2 and downregulation of NOTCH2 and NF2L2 under hypoxic conditions) but lacks a discussion on the functional implications of these changes. Expanding on how these specific gene regulations contribute to the pathology of glioblastoma or resistance mechanisms could provide valuable insights into the potential therapeutic impact of MRS1220.
3. The validation of RNAseq results through RTqPCR for three genes (TRIB2, NOTCH2 and NF2L2) is mentioned, which is good practice. However, details about the selection criteria for these genes for validation (beyond being altered under hypoxic conditions) would add depth to the methodology. Were these genes selected based on their fold change, significance level, or known biological relevance to glioblastoma?
4. In paragraph 2.2. Given the relatively small number of genes affected by MRS1220 treatment under hypoxia, the paragraph could expand on the significance of these changes. Are the genes that were downregulated or upregulated by MRS1220 under hypoxic conditions known to play critical roles in cancer progression, chemoresistance, or tumor microenvironment adaptation? Highlighting the impact of these changes could strengthen the argument for MRS1220's potential therapeutic value.
5. In paragraph 2.3. Mentioning how targeting these genes with MRS1220 compares to current glioblastoma treatments could provide context regarding the therapeutic potential of MRS1220. Are there existing therapies that target these pathways, and if so, how does MRS1220's approach differ or improve upon these strategies?
Comments on the Quality of English LanguageThe paper's English quality is notably high, reflecting a scholarly and professional approach suitable for its topic. It successfully conveys its research findings, employing precise scientific terminology and structured presentation. The authors clearly present their analysis and arguments, making the complex study of A3AR antagonism, chemoresistance, and gene expression in glioblastoma stem-like cells understandable.
Author Response
We greatly appreciate the comments provided. We make the suggested modifications and add the answers point by point.
- additional validation in other GB cell lines are needed.
R: We highly value this review. We carried out new validations in primary cultures of GSCs, which were added to the manuscript. Figure 1 supplementary
- The paragraph 2.1 briefly mentions the outcome of the treatment (e.g., upregulation of TRIB2 and downregulation of NOTCH2 and NF2L2 under hypoxic conditions) but lacks a discussion on the functional implications of these changes. Expanding on how these specific gene regulations contribute to the pathology of glioblastoma or resistance mechanisms could provide valuable insights into the potential therapeutic impact of MRS1220.
R: Thank you very much for the comment. In the discussion, we added information about the genes NOTCH2 and NFE2L2, which had yet to mention their functional implications.
- The validation of RNAseq results through RTqPCR for three genes (TRIB2, NOTCH2 and NF2L2) is mentioned, which is good practice. However, details about the selection criteria for these genes for validation (beyond being altered under hypoxic conditions) would add depth to the methodology. Were these genes selected based on their fold change, significance level, or known biological relevance to glioblastoma?
R: Indeed, to provide further depth to the methodology, we added the selection criteria of these genes for validation to the text in the results section.
- In paragraph 2.2. Given the relatively small number of genes affected by MRS1220 treatment under hypoxia, the paragraph could expand on the significance of these changes. Are the genes that were downregulated or upregulated by MRS1220 under hypoxic conditions known to play critical roles in cancer progression, chemoresistance, or tumor microenvironment adaptation? Highlighting the impact of these changes could strengthen the argument for MRS1220's potential therapeutic value.
R: Thank you very much for this comment. We added a table with molecular functions, biological processes, and pathways related to down-regulated and up-regulated genes after MRS1220 treatment under hypoxic conditions.
- In paragraph 2.3. Mentioning how targeting these genes with MRS1220 compares to current glioblastoma treatments could provide context regarding the therapeutic potential of MRS1220. Are there existing therapies that target these pathways, and if so, how does MRS1220's approach differ or improve upon these strategies?
R: We added to the discussion a paragraph explaining a possible adjuvant action of MRS1220 with Temozolomide.

Round 2
Reviewer 1 Report
Comments and Suggestions for Authors
The authors have made revisions to the manuscript and addressed the reviewers' comments, rendering it suitable for publication in its current form. The study sheds light on the upregulation of three genes post-treatment with MRS1220, indicating their potential as targets of this gene. While the conclusions are straightforward and could serve as informative references for more detailed investigations, it's worth noting that the transcriptomics analysis is somewhat limited due to the use of duplicate rather than triplicate samples, impacting the statistical strength, especially in comparisons involving groups of only two samples.
Moreover, the experimental design is relatively simplistic, and the results heavily rely on literature data without experimental validation. However, if the primary goal of the manuscript is to provide a preliminary direction for further research, I am inclined to support its publication. In summary, while the manuscript may lack some methodological robustness, its contribution lies in offering insights for future investigations. Thus, if the aim is to stimulate further exploration in the field, I recommend considering it for publication.
While I acknowledge these limitations, if the primary aim of the manuscript is to stimulate further research and serve as a starting point for more in-depth investigations, then it holds merit for publication.
Author Response
Thank you very much for the reviews; we want this work to be a starting point for more profound research.
Reviewer 2 Report
Comments and Suggestions for Authors
The authors provided answers, but the text modifications on the indicated lines are not marked yellow, making it difficult for me to assess them.
Author Response
We greatly appreciate the comments provided. We make the suggested modifications and add the answers point by point.
- There is a possibility that targeting A3AR components may have unintended consequences on other cellular processes.
R: Thank you very much for the valuable comment, we added this point to the discussion. Line 426 in yellow
- What are the possible alternative therapies for treating glioblastoma apart from targeting the components of GSCs?
R: We add to the discussion different therapies that can be used for glioblastoma and GSCs. Line 285-300 in yellow
- There is a possibility that other factors besides GSCs contribute to chemoresistance in glioblastoma.
R: Indeed, there are other chemoresistance mechanisms apart from GSCs, and we added a paragraph to the discussion mentioning these mechanisms. Line 283-285 in yellow

Reviewer 3 Report
Comments and Suggestions for Authors
have addressed all issues.
Author Response
We greatly appreciate the reviews made.
Round 3
Reviewer 2 Report
Comments and Suggestions for Authors
The authors addressed my concerns and revised the Discussion section. I find their manuscript acceptable for publication.